# The quantum cartpole: A benchmark environment for non-linear reinforcement learning

Kai Meinerz[1⋆], Simon Trebst[1], Mark Rudner[2] and Evert van Nieuwenburg[3]

**1** Institute for Theoretical Physics, University of Cologne, 50937 Cologne, Germany
**2** Department of Physics, University of Washington, Seattle, Washington 98195-1560, USA
**3** Lorentz Institute and Leiden Institute of Advanced Computer Science,
Leiden University, P.O. Box 9506, 2300 RA Leiden, The Netherlands

⋆ kmeinerz@thp.uni-koeln.de

## Abstract

**Feedback-based control is the de-facto standard when it comes to controlling classical stochastic systems and processes. However, standard feedback-based control methods are challenged by quantum systems due to measurement induced backaction and partial observability. Here we remedy this by using weak quantum measurements and model-free reinforcement learning agents to perform quantum control. By comparing control algorithms with and without state estimators to stabilize a quantum particle in an unstable state near a local potential energy maximum, we show how a trade-off between state estimation and controllability arises. For the scenario where the classical analogue is highly nonlinear, the reinforcement learned controller has an advantage over the standard controller. Additionally, we demonstrate the feasibility of using transfer learning to develop a quantum control agent trained via reinforcement learning on a classical surrogate of the quantum control problem. Finally, we present results showing how the reinforcement learning control strategy differs from the classical controller in the non-linear scenarios.**

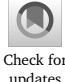

# 1   Introduction

Feedback-based control is essential in many different industries and domains. Stabilizing temperatures, chemical reactions, robotics and even biomedical devices are all possible by continuously adjusting the system inputs based on real-time feedback. Applications of this type of control to quantum systems has not yet reached this level of maturity, though several other quantum optimal control methods, such as GRAPE and CRAB [1–3] have gained more widespread adoption. A key issue with feedback-based control for quantum systems is that measurements of the quantum system cause measurement back-action [4–6] , limiting the amount of information that can be obtained. In many cases that renders standard optimal control techniques inapplicable or infeasible, either because they require a model of the quantum system or because they need gradients that require many measurements to estimate.

In this work we discuss a simple quantum problem for benchmarking feedback-based control, building on the *quantum cartpole* [7]. This control problem is based on the classical cartpole problem, which has become the de-facto standard benchmark for reinforcement learning controllers. We adapt the quantum cartpole problem to include explicit weak measurements of position and momentum, and use a continuous control parameter for feedback. In addition, we introduce a classical surrogate for this quantum problem by mimicking the measurement feedback on the system and the uncertainty in the measurements via noise. For this classical model, we investigate a standard optimal control algorithm – linear quadratic Gaussian control (LQGC), and show that this same controller can also control the quantum system based on weak measurement inputs. In regimes where the standard optimal controller struggles, e.g. for more non-linear systems or in scenarios where a noise characterization is infeasible, we demonstrate that deep reinforcement learning [8] remains a valid option for achieving control.

Deep reinforcement learning (RL) has been demonstrated to be a useful tool for control in several previous works [9–14], starting with [15]. It provides a general approach to devising control strategies in cases where a model of the system's dynamics is incomplete, or where other properties such as the noise model are unknown.

This work hence fits into the more general context of machine learning applications to quantum systems [16]. Some of those have used RL for feedback-based control, such as [7, 17,18]. An interesting recent work has also explored the use of weak nonlinear measurements as a way to compensate for purely linear controllers [19].

## 2   The quantum cartpole

The quantum system we consider in this work is derived from the well-known classical cartpole problem [20]. In it, a cart rolls on a flat one-dimensional track, and a force must be applied in either direction in order to keep a pole, hinged to the middle of the cart, upright. At every timestep, the system is described by a vector $s_t = (x_t, \dot{x}_t, \theta_t, \dot{\theta}_t)$ containing the instantaneous position $x_t$ and pole angle $\theta_t$ and their time derivatives. When the angle $\theta$ exceeds a threshold angle $\theta_{\text{th}}$, the failure condition is met and the controller failed to stabilize the system. The time step at which this failure condition is met is labelled the termination time $t_{\text{termination}}$.

A related but slightly simpler control problem is that of a particle sliding off of a hill needing to be pushed back up, for which the state vector is simply $s_t = (x_t, \dot{x}_t)$ and for which the failure condition is when the particle slides beyond a certain distance $x_{\text{th}}$, e.g., $|x| > |x_{\text{th}}|$. Although this system is an inverted pendulum, the quantum version of this problem has been dubbed the *quantum cartpole* [4,7], in which a free particle [initialized as a Gaussian wavepacket $\psi(x)$] is centered on an inverted potential $V(\hat{x})$ (for which we will discuss several choices later). The system undergoes unitary dynamics governed by the Hamiltonian

$$\hat{H} = \frac{\hat{p}^2}{2m} + V(\hat{x}),\tag{1}$$

and the failure condition is now set by *at least* 50% of the wavepacket's probability density extending beyond $x_{\text{th}}$. The state vector of this system is the wavefunction $\psi(x)$, though a controller will not have access to it. Instead, the controller has access to the results of weak measurements on the system, based on which the controller must decide to apply a particular unitary 'kick' $u_F$ to the system. This control force is realized through a momentum shift operator, i.e. $u_F = e^{-iF\hat{x}}$. Differently from [7], we will allow the controllers to choose a continuous $F$ (with bounded strength $|F| < |F_{\text{max}}|$) and we will provide the controller access to data from repeated discrete weak measurements of position and momentum (rather than continuous measurements of position only).

The full dynamics of the problem are hence as follows. First, a Gaussian wavepacket is initialized at the top of the inverted potential. The wavepacket has a width set by $\sigma = 1.0$,

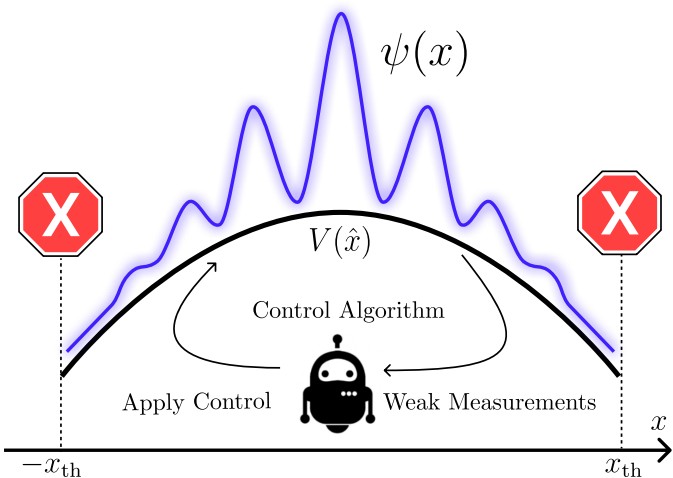

Figure 1: **The quantum cartpole setup.** A wavepacket $\psi(x)$ is placed on top of an inverted potential $V(\hat{x})$, and a controller must ensure that at least 50% of $|\psi(x)|^2$ stays within the interval $[-x_{\text{th}}, x_{\text{th}}]$. As input, the control algorithm receives weak measurement outcomes, and as output it sets the strength of a unitary control force that is applied to the wavepacket.

and has an initial momentum chosen randomly from a zero mean uniform distribution of width $\sigma_p = \sqrt{\langle p_{\text{init}}^2 \rangle} = 0.1$, where all units are set to 1. A more detailed description of the parameters and some comments regarding the units can be found in App. A.1. From then on, in every time step $\Delta t$:

(i) The system evolves unitarily under Hamiltonian (1) for a time $\Delta t$;

(ii) weak position and momentum measurements are performed, sequentially, and are reported to the controller (more details below);

(iii) the unitary operator $u_F$ is applied to the system, with $F$ chosen by the controller.

This loop is the core of the dynamics, and runs until the failure condition is met.

To investigate a trade-off between different timescales of the dynamics and control, we introduce a variable $N_{\text{meas}}$ representing the number of times the dynamics loop runs until the force $F$ can be changed. This is shown schematically in Fig. 2. In each repetition we perform a weak measurement of the system's position and momentum [21], and average them into $\overline{x}_{\text{est}}$ and $\overline{p}_{\text{est}}$, which are both passed to the control algorithm. Notice that this is not identical to performing an $N_{\text{meas}}$ times stronger weak measurement, as between every single measurement the wave function evolves in time over the duration $\Delta t$, which is kept constant independent from $N_{\text{meas}}$.

The pre-processing step of averaging the weak measurement results (rather than providing the measurements directly and using, e.g., a controller with memory), is inspired by the frame-stacking technique used in reinforcement learning [22] and has the goal of getting better estimates of the position and momentum, and thereby preventing the controllers from applying too strong or too weak forces. The weak measurements performed at each step are essential for control. As a specific implementation, one can consider these measurements to be performed by coupling the quantum cartpole system with an ancilla system for a short period that is then projectively measured (see App. A.2 for a detailed derivation). Without the weak position measurement, the wavepacket would continue delocalizing irrespective of the unitary force $u_F$. Finally, we mention that we have implemented this control problem as an OpenAI Gymnasium environment [23], making it suitable for reinforcement learning control.

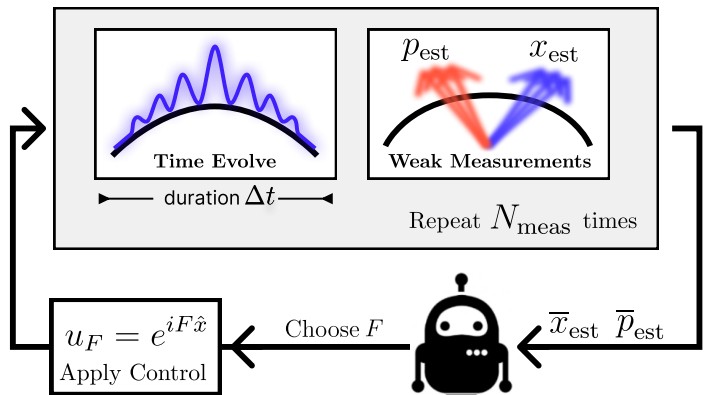

Figure 2: **Scheme of the dynamics** of the quantum cartpole problem. In every time step $\Delta t$ a force $F$ is applied, the wavefunction is evolved in time and a weak position and momentum measurement are performed. These steps are repeated $N_{\text{meas}}$ times. Afterwards, the mean values of the weak measurements $\overline{x}_{\text{est}}$ and $\overline{p}_{\text{est}}$ are passed to a control algorithm, which will decide on a value for the stabilizing force $F$.

## 2.1 Classical surrogate of the quantum cartpole

To be able to compare the aspects of controlling this quantum system versus a classical system, we introduce a noisy (stochastic) classical version whose noise properties are tuned to mimic the uncertainty of weak measurements. For the rest of this work, the noisy classical system refers to the following implementation rather than the original classical cartpole.

This classical system is a linear stochastic system, describing a classical particle on an inverted potential, given by

$$s_{t+1} = As_t + Bu_t + w_t\,, \tag{2}$$

$$y_t = Cs_t + v_t\,, \tag{3}$$

where $s_t = (x_t, \dot{x}_t)$ is the state vector introduced above, $y_t$ describes the results of measurements (and hence the inputs to the controller), and $u_t$ is the control vector set by the control algorithm, containing the force applied to the system (playing a role analogous to $u_F$ in the quantum problem). The matrices $A$, $B$, and $C$ describe the dynamics and measurements (see App. A.1), and the vectors $w_t \sim \mathcal{N}(0, \sigma_{\text{dyn}})$ and $v_t \sim \mathcal{N}(0, \sigma_{\text{meas}})$ describe noise on the dynamics and on the measurements, respectively. That is, $w_t$ and $v_t$ are normally-distributed random variables with zero mean and standard deviations $\sigma_{\text{dyn}}$ and $\sigma_{\text{meas}}$, respectively. The uncertainty coming from the weak measurements is reflected in $v_t$, and the measurement backaction is reflected through noise on the state, $w_t$. Because the backaction depends on the measurement outcome, the noise models for $v_t$ and $w_t$ are correlated (see App. A.3). Also in this case, the particle (now a point-mass) is initialized on top of the inverted potential with a random momentum chosen from a uniform distribution of width 0.1.

## 3 Control algorithms

Now that we have a quantum problem and a classical problem that mimics it, we turn our attention towards two possible control strategies. In particular, we ask how well an optimal controller for the classical variant performs on the quantum version, and whether a reinforcement learning controller can go beyond such optimal control in scenarios where the latter struggles.

## 3.1 Linear quadratic Gaussian control

The well-known linear quadratic Gaussian control (LQGC) algorithm is a classical control algorithm that is known to optimally control a *linear* system subjected to Gaussian noise [24,25]. The algorithm itself consists of two parts: the Kalman filter [26,27] (the estimator) and the linear quadratic regulator (LQR) (the controller) [28]. The latter assumes that we can apply linear control, $u_t = -K_{\text{LQR}}x_t$ in Eq. (2), and provides $K_{LQR}$ by minimizing a quadratic cost function $J$ (see App. A.4.4). The performance of this controller (without the Kalman filter) on the classical problem with a *quadratic* inverted potential is shown in Fig. 3a for various numbers of measurements and for several values of noise, where the performance is measured in terms of $t_{\text{termination}}$, which is the average number of time steps $\Delta t$ until the termination condition is met.[1] There is an intuitive trade-off where more measurements allow for better control (averaging out the noise), but where too many measurements is detrimental since they take too much time. In that time, the system either reaches the failure condition or goes beyond the point where control is possible (e.g., because a control force $|F| > |F_{\text{max}}|$ would be required).

---

[1]In the absence of noise, this system can be stabilized indefinitely.

Better performance can be achieved by incorporating an estimator such as the Kalman filter into the feedback protocol. The Kalman filter provides an estimate of the system's state $\hat{s}_t$, by *using a model* of the underlying dynamics to calculate the most probable state of the system based on the measurement $y_{t-1}$, the previous state estimation $\hat{s}_{t-1}$. Appendix A.4 describes how this is done for a linear system in more detail. Figure 3b shows that when we use the Kalman filter, the trade-off disappears entirely and a single measurement provides the best result. This is true for Markovian systems, for which the current state of the system only depends on the previous state (and not further history), so that knowing the previous state provides all information to provide a good estimate of the current state.

## 3.2 Reinforcement learning control

Because we will move away from the scenario where LQGC is designed to work, we turn our attention to reinforcement learned control. A possible advantage of such controllers is that they can learn control without having access to a model and without access to the noise model (i.e., without explicitly making use of Eqs. (3)). Hence, being *model free* and relying only on measurement results as input, the agents we study can be applied either to the control of quantum or classical systems (though performance and optimal parameters may be different for the two cases). A thorough introduction to reinforcement learning control can be found in [8], and we mention specifically here that reinforcement learning agents are capable of learning the LQR algorithm [29].

Inspired by LQGC, rather than training a single agent to stabilize the quantum cartpole, we train two distinct agents: one for determining the control force (the reinforcement learned controller, RLC) and another responsible for state estimation (the reinforcement learned estimator, RLE). Both agents are trained using a stochastic on-policy training algorithm called the *proximal policy optimization* (PPO) algorithm [30], and both use continuous input and output spaces. Detailed of the training process, a short description of the PPO algorithm, and the parameters used are listed in Appendix A.5.

The first reinforcement learning agent – the reinforcement learning controller (RLC) – is trained with the goal of providing the control input based on the raw measurement inputs. The input to the agent is hence directly $\overline{x}_{\text{est}}$ and $\overline{p}_{\text{est}}$. As output the agent returns a controlling force $u_F$ from the range $[-F_{\max}, F_{\max}]$ for the next time step. The reward is $-1$ if the control fails (i.e., the wavefunction moves outside the boundaries), and 0 every time step otherwise. Testing the RLC on a classical system with inverse *quadratic* potential in Fig. 3a, we see that it has the same trade-off as the LQR controller, but performs slightly better due to difference in the objective in both algorithms. The LQR minimizes the quadratic cost during the run, whereas the RLC aims to avoid the worst case of the wavefunction being pushed out of the threshold.

The second agent – reinforcement learned estimator (RLE) – the is trained with the goal of replacing the Kalman filter. To do so, we provide it with the previous state estimate $\hat{s}_{t-1}$, the mean of the $N_{\text{meas}}$ measurements taken at timestep $t$ and the last control value $u_t$. Compared to the Kalman filter, however, the agent has no knowledge about the noise covariance nor of the system's equations of motion. As output the agent returns the predicted change of the state $\Delta\hat{s}_t$, so that $\hat{s}_{t+1} = s_t + \Delta\hat{s}_t$. The goal of the training is to minimize the squared prediction error $e_t^2$, where $e_t = y_t - CAs_{t-1}$ (see App. A.4), by providing a reward $r_t = -e_t^2$ in each time step. For the purpose of training, the controller is replaced by a simple random controller (choosing a random force every time step), since the estimation task does not depend on the actual control strategy.

Putting the estimator to the test in combination with the RLC on the classical surrogate system with inverted quadratic potential is shown in Fig. 3b, where, like the LQGC, the performance always reaches the maximum for $N_{\text{meas}} = 1$, independent of the noise level. Overall,

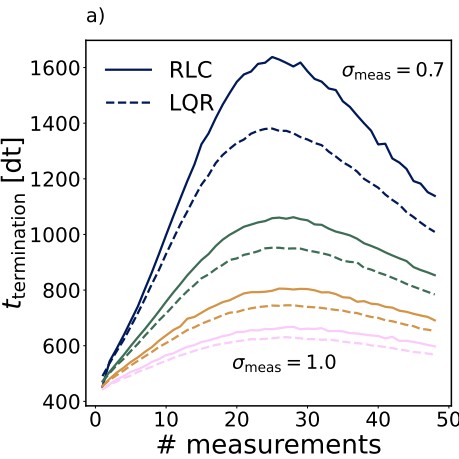
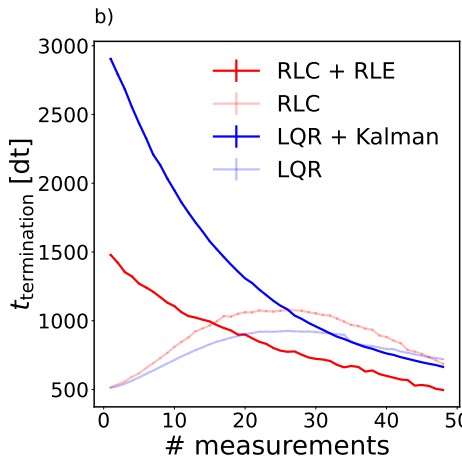

Figure 3: **Controller performance for noisy classical cartpole** (with inverted quadratic potential). Plot a) compares the RLC against the LQR for different levels of measurement noise $\sigma_{\mathrm{meas}}$ (see Eqs 3), showing a comparable increase and decrease in performance with increasing number of measurements. The RLC reaches a higher maximum compared to the LQR. In b) the noise level is fixed at $\sigma_{\mathrm{meas}} = 0.8$ and state estimators in form of the Kalman Filter and RLE are added to the comparison, shifting the peak performance to a single measurement.

the performance of the reinforcement learned estimator is below that of the Kalman filter, indicating that learning the state estimator is more challenging than learning the controller[2] and making LQGC the better choice if the system is linear and a noise model is available.

## 4 Controlling the quantum cartpole

We now turn our attention to the quantum version of the problem. In Fig. 4 we show how a reinforcement learning controller, trained on the quantum system, performs in this scenario (still with a quadratic inverted potential), once without the estimator (panel a) and then with (panel b). Here, too, the trade-off between more measurements for more information versus latency is apparent if only the controller is used. Panel a also shows that control quickly turns infeasible in the weak measurement regime (corresponding to the top of the panel), where the trade-off fades out, but still indicates a finite number of measurements remains optimal. Like in the classical case, using the estimator in addition makes it such that the optimum in $N_{\mathrm{meas}}$ shifts to a single measurement as seen in Fig. 3b.

### 4.1 Potential variations

To comprehensively explore the capabilities of our controllers, we consider four possible combinations of controllers (LQR and the RLC) and estimators (Kalman filter and the RLE). We explore how these combinations perform as a function of different numbers of measurements, evaluating them based on the average time $t_{\mathrm{termination}}$ (calculated over $10^5$ runs). To check the validity and usefulness of the LQGC versus the reinforcement learning controller, we investigate different potentials that make the system non-linear. The potentials that we study are depicted in Fig. 5, which next to the quadratic inverted potential shows two more:

---

[2]The training of the estimator agent converged, so the difference in performance is not due to training. More likely is that a more complex agent would be able to learn the system's dynamics and perform better estimation, compared to the standard PPO agent we chose.

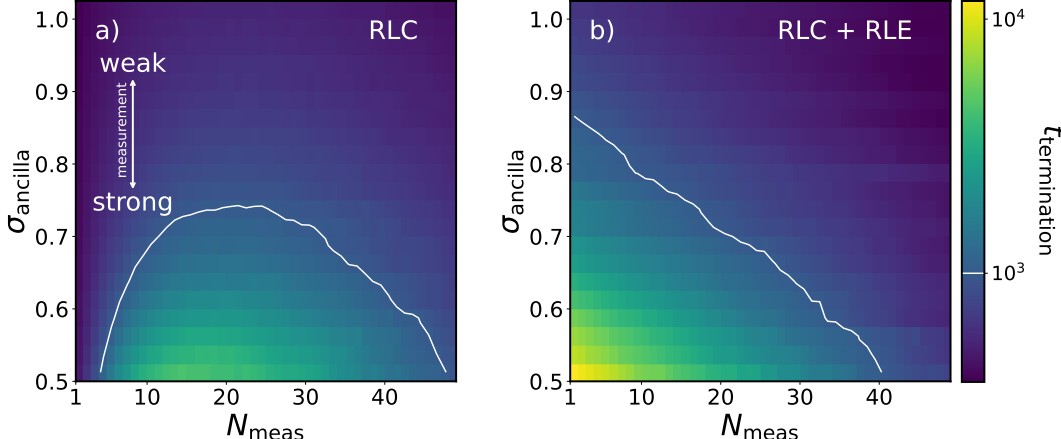

Figure 4: **Controller performance for quantum cartpole** with an inverted quadratic potential. Shown is the performance of various controllers (measured in the termination time $t_{\text{termination}}$) indicating a trade-off between the number of measurements $N_{\text{meas}}$ and the strength of the measurement, indicated by the width $\sigma_{\text{ancilla}}$ of the measurement wavefunction (see Appendix A.2 for details). Larger $\sigma_{\text{ancilla}}$ corresponds to weaker measurements. Panels a) and b) show RLC and RLC + RLE as controllers, respectively, with the heat maps showing the average termination time $t_{\text{termination}}$ on a logarithmic scale and the white line indicating $t_{\text{termination}} = 10^3$.

  (i)  A cosine potential $V(\hat{x}) = k_1 \left(\cos(\pi\hat{x}/k_2) - 1\right)$, and

 (ii)  An inverted quartic potential $V(\hat{x}) = -k\hat{x}^4$.

The values for $k_1, k_2$ and $k$ are listed in Appendix A.1, which also shows the performance of the controllers on the classical system with these potentials. For the RL based controllers, we trained multiple agents and averaged the results of the 10 best performing ones and presenting their performance relative to the LQGC performance, to highlight the improvement of the performance, independent from the concrete performance, which depends on the underlying potential.

For the quadratic inverted potential the combination of reinforcement learning controller (RLC) and the Kalman filter performs as well as the LQGC algorithm (see panel a of Fig. 6). This is a notable result, because the LQGC is not a guaranteed optimal controller in the *quantum* environment. However, our findings are consistent with those presented for discrete control of the quantum cartpole [7]. At the same time, we note that the reinforcement learned estimator (RLE) struggles to match the performance of the Kalman filter up to about $N_{\text{meas}} \sim 40$, and that does not remove the trade-off behavior discussed previously. For larger $N_{\text{meas}}$ the RLE seems to be able to match the Kalman performance.

Now going to a nonlinear system, starting with the cosine potential Fig. 6b, the overall performances are similar to the linear system. It is notable to see that the combination RLC + Kalman is now able to gain a notable advantage over the LQGC, increasing the performance by $\sim 10\%$ for a single measurement $N_{\text{meas}} = 1$. Similarly, the controllers involving RLE were able to close the performance gap to the LQGC by small margins, but remain far behind.

It is for the quartic potential that the advantage of using RL becomes really evident. Here the RLC + Kalman controller is able to achieve an increase in performance of $\sim 60\%$ compared to the LQGC. At the same, we also see that both the RLC + RLE and the LQR + RLE controllers are both able to also achieve a performance advantage over the LQGC and narrow the gap with the RLC + Kalman controller, indicating that the LQR algorithm is the main bottleneck.

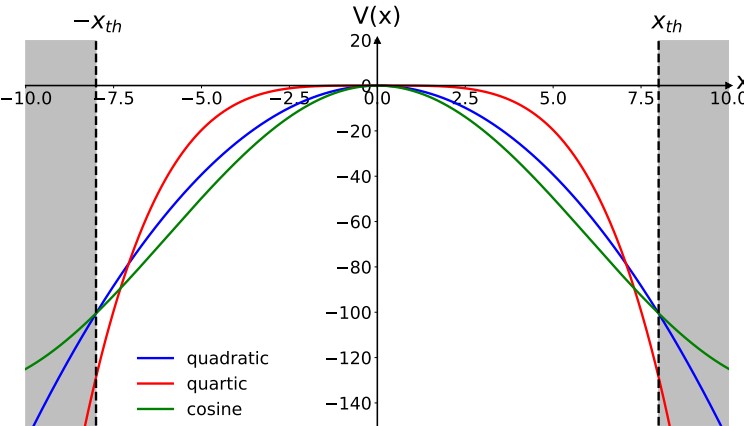

Figure 5: **The different potentials** used for the quantum cartpole problem. The cosine and quartic (red and green) potentials are used to demonstrate behavior on nonlinear systems, while the quadratic (blue) potential is used for a linear system.

On a broader level, we expect that for even more non-linear problems reinforcement learning control indeed becomes the go-to choice instead of LQGC.

## 4.2 Transfer learning

Finally, we consider a *transfer learning* scenario in which we train a reinforcement learning agent on the classical surrogate, and then apply it to control the quantum system. The results of these comparisons for the different combinations of controllers and estimators are shown in Fig. 6. Interestingly, RL agents trained on the classical system perform almost identically to those trained on the quantum system (comparing the 'transfer' controllers with their counterparts). This suggests that training on a classical surrogate model for controlling the quantum cartpole is indeed a viable strategy.

## 4.3 Controller characteristics

To further elucidate the control strategies used by the control algorithms, we investigate the resulting distributions of position $\langle x \rangle$ and momentum $\langle p \rangle$ expectation values, shown in Fig. 7. For this we compare the LQGC and the RLC + RLE controllers on the three different potentials. The distributions were taken by collecting the position and momentum of the wavefunction over $10^6$ time steps. In order to avoid measurement artifacts from the initialization of the wavefuntions, only data from $t = 300$ and onwards were taken.

Looking first at the LQGC, it can be observed that the distributions for all potentials are symmetrical and centered around 0, showing that the controller aims to stabilize the wavefunction at the centers of the potentials. When comparing the quadratic and cosine potentials, it is notable that the cosine potential has a wider distribution due to the fact that it starts to flatten out near the threshold. It appears that this allows the controller to stabilize it closer to the threshold for longer durations. In contrast, the quartic potential has the sharpest distribution, suggesting that it is unable to recover the wavefunction when it is close to the thresholds.

Looking at the distributions of the full reinforcement learning control algorithm (RLE + RLC) one notable disparity is observed. The distributions are neither centered around 0 nor are they symmetric. This is attributed to the training process, where the agent can develop a bias for stabilizing the wavefunction at a particular point. This is particularly evident in



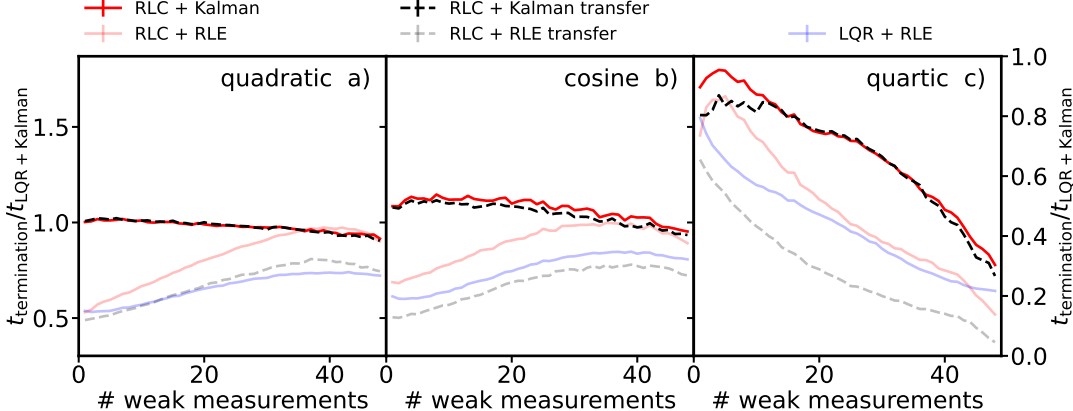

Figure 6: **Benchmarks of the various controllers**, on the **quantum** system depending on the number of measurements for the input. We showcase the performance of the Kalman Filter + LQR (blue) and the pure reinforcement learning controller (light red), as well as the mix of those controllers with Kalman Filter + RLC (red) and LQR + RLE (light blue). Additionally, transferred RLC + Kalman Filter and RLC + RLE (black) are presented, which were trained on the classical system and then applied on the quantum system. The performance is showcased as ratio of the average termination time between a selected controller and the Kalman + LQR controller. Each plot represents a different potential, the first being the **quadratic** potential, followed by the **cosine** potential and **quartic** potential.

the quadratic and quartic potentials, where the wavefunction is stabilized left and right of the center.

For the cosine potential, the distribution of the average position with RLC + RLE significantly differs from that of the LQGC. Instead of a clear peak in the distribution a broader plateau appears, indicating that the controller has learned to balance the wavefunction on the side of the potential rather than the center.

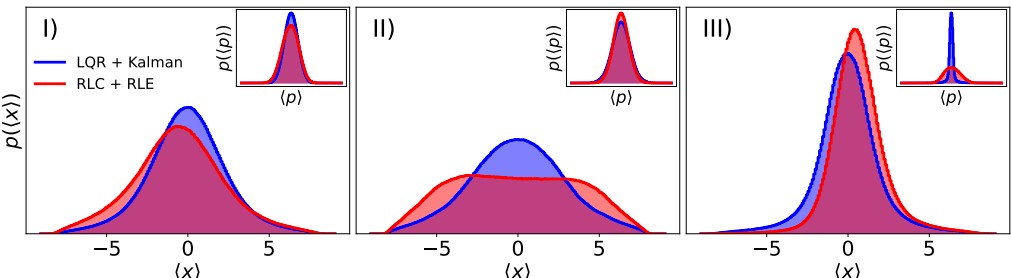

Figure 7: **Distributions of the position and momentum of the stabilized wavepacket** on the quantum system for the three different potentials: (I) **quadratic**, (II) **cosine**, (III) **quartic**. The position $\langle x \rangle$ of the wavefunction was tracked over $10^6$ timesteps for the **LQR + Kalman** controller (blue) as well as the **RLC + RLE** (red) controller and converted to a histogram of the position probability $p(\langle x \rangle)$. This is also showcased for the momentum $\langle p \rangle$ in the top right insets.

## 5 Conclusions and Outlook

Standard classical feedback-based control algorithms are challenged by quantum systems, due to their intrinsic measurement induced backaction and partial observability. However, for quantum systems with observables that can approximately be described by linear stochastic equations, we found that the classical LQGC algorithm performs well if full knowledge of the model as well as its noise characteristics are available. If such knowledge is not available, or if the system is non-linear, the classical controller is found to struggle. In scenarios where knowledge of the model and the noise are unavailable (as is often of the case for complex experiments), and/or where the system is non-linear, we showed that a model-free reinforcement learned controller outperforms the LQGC algorithm.

To fairly demonstrate the advantage of our reinforcement learning controller, we constructed a surrogate model in the form of a classical stochastic system, whose properties are designed to closely mimic the measurement-induced backaction of the quantum cartpole. Using this classical surrogate, we demonstrate that transfer learning is feasible by training the RL agent on this classical model and applying it to controlling the true quantum system. This opens up the possibility of more efficient training methods.

Both the LQGC algorithm as well as our reinforcement learned controller are composed of a separate estimator and a controller, and we show that without the estimator a trade-off between the number of measurements and the controllability exists. Including the estimators removes this trade-off, allowing for optimized control with just a single (weak) measurement. For the LQGC algorithm, estimating the state (with the Kalman filter) again requires knowledge of the model and the noise characteristics. The reinforcement learned estimator struggles to match the performance of the Kalman filter, but still ensures controllability with single measurements. Frame-stacking techniques improve upon this further.

Finally, we found that when analyzing the control strategies for the linear system, the reinforcement learning controller learns a strategy similar to the optimal strategy of the LQGC. For the non-linear case, the reinforcement learning controller manages to outperform the LQGC algorithm by stabilizing the system with a different control strategy that allows for a broader distribution of the system's momentum.

It would be an interesting future direction to focus on making the RL agent more autonomous, by having it choose when and how strong to perform the weak measurements. This would result in an adaptive algorithm that may learn to only measure when necessary. Similarly, instead of frame-stacking the agent could learn to use the raw weak measurement outputs instead, instead of only their average. Finally, future work could focus on extending the system in interesting directions such as time-varying potentials, non-Markovian noise, or interacting systems.

## Acknowledgments

K. M. thanks the Niels Bohr Institute for hospitality during the initial stages of this work. S.T. thanks A. Sengupta for inspiring discussions.

**Funding information** We acknowledge partial funding through the Deutsche Forschungsgemeinschaft (DFG, German Research Foundation) – Projektnummer 277101999 – TRR 183 (projects A01, B01). The numerical simulations have been performed on the JUWELS Booster at FZ Julich. This work also recieved funding from the Dutch National Growth Fund (NGF) as part of the Quantum Delta NL programme. M.R. acknowledges the Brown Investigator Award, a program of the Brown Science Foundation, the University of Washington College of Arts and Sciences, and the Kenneth K. Young Memorial Professorship for support.

**Data availability**    The reinforcement learning environment [31] (in the form of an OpenAI Gymnasium) as well as the code and all configuration data used for obtaining the benchmarks in this paper [32], are made available.

Table 1: **Parametrization of the environment** used for the benchmarks. Including the parameters of the potentials, the time evolution, and the weak measurements. All units are set to 1.

| System Parameters | | | |
|---|---|---|---|
| parameter | quadratic | cosine | quartic |
| $k$ | $\pi$ | $67$ | $\frac{\pi}{100}$ |
| $\Delta t$ | $\frac{0.01}{\pi}$ | $\frac{0.01}{\pi}$ | $\frac{0.01}{\pi}$ |
| $m$ | $\frac{1}{\pi}$ | $\frac{1}{\pi}$ | $\frac{1}{\pi}$ |
| $\lambda$ | $0.05$ | $0.05$ | $0.05$ |
| $\sigma_{\text{system}}$ | $1.0$ | $1.0$ | $1.0$ |
| $\sigma_{\text{ancilla}}$ | $0.7$ | $0.7$ | $0.7$ |
| $\langle p^2 \rangle_{\text{init}}$ | $0.1$ | $0.1$ | $0.1$ |
| $x_{\text{th}}$ | $8$ | $8$ | $8$ |
| $F_{\text{max}}$ | $8\pi$ | $8\pi$ | $8\pi$ |

# A    Appendix

## A.1    System parameters

For the simulation in order to make fair comparison between the different potential, we have used the same initialization parameters on the on all three potentials, with the exception of the potential constant $k$.

## A.2    Weak measurement

We want to perform the weak measurement on the quantum state $|\Psi\rangle = |\psi\rangle \otimes |\phi\rangle$, where $|\psi\rangle$ is the system state and $|\phi\rangle$ the ancilla state. The two systems interact via the Hamiltonian $H_{\text{int}} = A \otimes p$. We assume that the interaction time $\delta t$ is small enough, thus the time evolution is dominated by the weak measurement. The time evolution can be written as $|\Psi'\rangle = U(t)|\psi\rangle \otimes |\phi\rangle$ with :

$$U(t) = \exp[-i\lambda H_{\text{int}}], \tag{A.1}$$

with $\lambda = s\delta t$ and $s$ being the interaction strength and $\Delta t$ the interaction time. Here $A$ is a Hermitian operator with the eigenstates $\alpha$, acting on the system quantum state $\psi$, and $p$ is the momentum operator acting on the ancilla state $\phi$.

Choosing the form $|\phi(q)\rangle = \frac{1}{(2\pi\sigma^2)^{(1/4)}} \int dq' \exp[-q'^2/(4\sigma^2)]|q'\rangle$ for the ancilla state, perform a projective measurement in the q state using $\prod_q = I \otimes |q\rangle\langle q|$. This leaves the state in the form $|\Psi_{q_m}\rangle = \frac{M_{q_m}|\Psi'\rangle}{\mathcal{N}}$, and returns the measured quantity $q_m$ and the Kraus operator

$$M_{q_m} = \frac{1}{(2\pi\sigma^2)^{(1/4)}} \int_{-\infty}^{\infty} d\alpha \, \exp[-(q_m - \lambda\alpha)^2/(4\sigma^2)]|\alpha\rangle\langle\alpha|, \tag{A.2}$$

which is a Gaussian weighted sum of projectors onto the eigenstates of $A$.

From the Kraus operator the probability density of the measurement follows as

$$P(q) = Tr\left[M(q)^\dagger M(q)|\psi\rangle\langle\psi|\right] \tag{A.3}$$

$$= \frac{1}{\sqrt{2\pi}\sigma}\int_{-\infty}^{\infty} d\alpha |\psi(\alpha)|^2 e^{-(\lambda\alpha-q)^2/(2\sigma^2)}. \tag{A.4}$$

Next we want to calculate the uncertainty of the measurement $\sigma_q = \langle q^2\rangle - \langle q\rangle^2$ using the probability density. Starting with the expectation value of $q$,

$$\langle q\rangle = \int_{-\infty}^{\infty} dq\, q P(q) \tag{A.5}$$

$$= \frac{1}{\sqrt{2\pi}\sigma}\int_{-\infty}^{\infty}\int_{-\infty}^{\infty} dq\, d\alpha\, q |\psi(\alpha)|^2 e^{-(\lambda\alpha-q)^2/(2\sigma^2)} \tag{A.6}$$

$$= \int_{-\infty}^{\infty} d\alpha |\psi(\alpha)|^2 \lambda\alpha \tag{A.7}$$

$$= \lambda\langle A\rangle, \tag{A.8}$$

one can see that the expectation value of the measurement is given by the expectation value of the operator $A$. If we assume that ancilla wavefunction is much broader than the system wavefunction, we can approximate $|\psi(\alpha)|^2$ with a delta function.

$$P(q) \approx \frac{1}{(2\pi\sigma^2)^{(1/2)}}\int_{-\infty}^{\infty} d\alpha\, \delta(\alpha'-\langle A\rangle)\exp[-(q-\lambda\alpha)^2/(2\sigma^2)]$$

$$= \frac{1}{(2\pi\sigma^2)^{(1/2)}}\exp[-(q-\lambda\langle A\rangle)^2/(2\sigma^2)].$$

This allows us to write the measurement result $q$ as a stochastic quantity

$$q = \lambda\langle A\rangle + \Delta W, \tag{A.9}$$

where $\Delta W$ is a zero-mean Gaussian random variable with a variance $\sigma^2$.

The same derivation can be done for $\langle q^2\rangle$.

$$\langle q^2\rangle = \int_{-\infty}^{\infty} dq\, q^2 P(q) \tag{A.10}$$

$$= \frac{1}{\sqrt{2\pi}\sigma}\int_{-\infty}^{\infty}\int_{-\infty}^{\infty} dq\, d\alpha\, q^2 |\psi(\alpha)|^2 e^{-(\lambda\alpha-q)^2/(2\sigma^2)} \tag{A.11}$$

$$= \int_{-\infty}^{\infty} d\alpha |\psi(\alpha)|^2 \left[2\sigma^2 + \lambda^2\alpha^2\right] \tag{A.12}$$

$$= 2\sigma^2 + \lambda^2\langle A^2\rangle. \tag{A.13}$$

Putting both together yields the uncertainty of the measurement

$$\sigma_q^2 = \lambda^2\langle\alpha^2\rangle + 2\sigma^2 - \lambda^2\langle\alpha\rangle^2 \tag{A.14}$$

$$= \lambda^2\sigma_\alpha^2 + 2\sigma^2. \tag{A.15}$$

From this we can see that in the case $\lambda = 1$ and $\sigma \to 0$, the uncertainty of the measurement reduces to the uncertainty of the system wavefunction, recovering strong measurement. This

also translates to the uncertainty relation assuming the additional measurement using the operator $B$ with the eigenstates $\beta$ and that $A$ & $B$ do not commute, giving us the relation

$$\sigma_q^2 \sigma_p^2 = \sigma_\alpha^2 \sigma_\beta^2 \geq \frac{1}{4}. \tag{A.16}$$

Looking also in the other case with $\lambda = 0$, we have no interaction between the system and ancilla wavefunction and the uncertainty of a measurement reduces to $\sigma_q^2 = 2\sigma^2$ the variance of the ancilla wavefunction.

### A.3 Noise determination of classical system

The measurement and system noise of the classical system are defined by

$$\mathbf{E}\left[\begin{pmatrix} w_t \\ v_t \end{pmatrix} \begin{pmatrix} w_{t'}^T & v_{t'}^T \end{pmatrix}\right] = \begin{pmatrix} Q & S \\ S^T & R \end{pmatrix} \delta_{tt'}, \tag{A.17}$$

where $Q, R$ are the covariance matrices of the system and measurement noise, respectively, and the $S$ is the cross-covariance matrix between those two. Those matrices are chosen to mimic the quantum system as closely as possible.

Based on Eq. A.9, the measurement noise covariance matrix follows directly as:

$$Q = \begin{bmatrix} \sigma_{\text{ancilla}}^2 & 0 \\ 0 & \sigma_{\text{ancilla}}^2 \end{bmatrix}. \tag{A.18}$$

The covariance matrix $R$ of the system noise and the cross-covariance matrix $S$ depend on the width $\sigma_{\text{sys}}$ of the system wavefunction, which is varying around a fixed value during a run. The fixed value also depends on the underlying potential, so that the noise matrices change depending on the potential. Because of that we numerically determined the values to be as close as possible to the weak measurement back actions. This is done by running the quantum system for a large number of steps and extracting the measurement and system noise added by the weak measurements over $10^6$ timesteps. The matrices are then given by:

$$R = \begin{bmatrix} \text{cov}(x,x) & \text{cov}(x,p) \\ \text{cov}(p,x) & \text{cov}(p,p) \end{bmatrix}, \qquad S = \begin{bmatrix} \text{cov}(x,x_{\text{meas}}) & \text{cov}(x,p_{\text{meas}}) \\ \text{cov}(p,x_{\text{meas}}) & \text{cov}(p,p_{\text{meas}}) \end{bmatrix}. \tag{A.19}$$

### A.4 Linear quadratic Gaussian controller LQGC

The LQGC is an optimal control algorithm for linear systems subject to Gauss noise. It is composed of the Kalman filter (or linear quadratic estimator LQE), which is a recursive estimator using a time series of measurements to approximate the unknown variables, and the linear quadratic regulator quadratic regulator (LQR), which converts the estimated values into an applicable force.

#### A.4.1 Kalman filter

Assume we have a linear system, which can be described by the equations

$$s_{t+1} = As_t + Bu_t + w_t,$$
$$y_t = Cs_t + v_t,$$

where $u_t$ is the known input at the timestep $t$ and $w_t.v_t$ are the process and measurement noise respectively. It is assumed that the noises can be described as zero meas Gaussian noises with the covariances $Q, R$.

In order to predict the next step, only the estimation from the previous timestep and the measurement from the current timestep are needed. The state of the filter can be described using the a posteriori state estimate mean at the time $t$, including measurement up to the time $t'$, $\hat{s}_{t|t'}$, and the a posteriori estimate covariance matrix, $P_{t|t'}$, which is used as a measure for the accuracy of the state estimation.

Firstly the Kalman filter predicts the next state of the estimation, by updating the last state as if process noise is applied

$$\hat{s}_{t+1|t} = A\hat{s}_{t|t} + Bu_t,$$
$$P_{t+1|t} = AP_{t|t}A^T + Q.$$

Next the state has to be corrected using the latest measurement $y_{t+1}$ by calculating the difference between the measurement and the optimal forecast of the estimated state:

$$\hat{z}_{t+1} = y_{t+1} - C\hat{s}_{t+1|t}, \tag{A.20}$$
$$S_{t+1} = CP_{t+1|t}C^T + R, \tag{A.21}$$

where $S_{t+1}$ is the covariance of $y_{t+1}$. From this the Kalman gain $K$ follows as

$$K_{k+1} = P_{t+1|t}C^T S_{k+1}^{-1}, \tag{A.22}$$

and the state estimation and covariance can be updated as

$$\hat{s}_{t+1|t+1} = \hat{x}_{t+1|t} + K_{k+1}\hat{z}_{k+1}, \tag{A.23}$$
$$P_{t+1|t+1} = (I - K_{t+1}C)P_{t+1|t}. \tag{A.24}$$

Based on the estimation we can now also define the prediction error

$$e_{t+1} = y_{t+1} - C\hat{s}_{t+1|t+1}. \tag{A.25}$$

### A.4.2  Kalman Filter with same time step correlated noise

Normally it is assumed that the linear system has uncorrelated process and measurement noise, but in the case of weak measurement, we have correlation between both noise types. To accommodate this correlation, we rewrite the system equation, to be of uncorrelated noise, following the derivation from [33]

Using an arbitrary matrix $T$, we transform the system to

$$\begin{aligned} s_{t+1} &= As_t + Bu_t + w_t + T[y_t - Cs_t - v_t] \\ &= (A - TC)s_t + Bu_t + w_t + Ty_t - Tv_t \\ &= A^* s_t + u_t^* - w_t^*, \end{aligned}$$

with the new transition matrix $A^* = (A - TC)$, new known input $u_t^* = Bu_t + Ty_t$ and the new noise $w_t^* = w_t + Tv_t$.

Now we choose the matrix T by setting the correlation between the new system noise and the measurement noise to zero

$$E[w_t^* v_t'] = E[[w_t + Tv_t]v_t'] = S - TR = 0, \tag{A.26}$$

which yields

$$T = SR^{-1}. \tag{A.27}$$

From this follows the covariance of the new process noise as:

$$Q^* = Q - SR^{-1}S'. \tag{A.28}$$

Using the new state equation and covariance, the Kalman gain can be calculated normally.

### A.4.3 Extended Kalman filter

In the case that the system is described by the non linear functions $f$ and $h$

$$s_{t+1} = f(s_t, u_t) + w_t,$$
$$y_t = h(s_t) + v_t,$$

it is necessary for the Kalman filter to linearize the current estimate and covariance. This model is called the extended Kalman Filter [33]. While the overall strategy of the filter stays the same, there are some differences. We start by linearizing the system, with

$$A(t) = \frac{\partial f}{\partial s}|_{\hat{s}_t, u_t}, \tag{A.29}$$

$$C(t) = \frac{\partial h}{\partial s}|_{\hat{s}_t}, \tag{A.30}$$

$$\tag{A.31}$$

where $A, C$ are now the Jacobians of their respective functions. From the extended Kalman functions follows from the same process as the standard Kalman:

$$\hat{s}_{t+1|t} = f(s_t, u_t),$$
$$P_{t+1|t} = AP_{t|t}A^T + Q,$$
$$\hat{z}_{t+1} = y_{t+1} - c(\hat{s}_{t+1|t}),$$
$$S_{t+1} = CP_{t+1|t}C^T + R,$$
$$K_{k+1} = P_{t+1|t}C^TS_{k+1}^{-1},$$
$$\hat{s}_{t+1|t+1} = \hat{s}_{t+1|t} + K_{k+1}\hat{z}_{k+1},$$
$$P_{t+1|t+1} = (I - K_{t+1}C)P_{t+1|t}.$$

### A.4.4 Linear quadratic regulator

The second part of the LQGC consists of the linear quadratic regulator LQR, which is defined for a linear system by

$$s_{t+1} = As_t + Bu_t. \tag{A.32}$$

In the case of inverted quadratic potential, we can use the A, B from the state equations, whereas for the inverse quartic potential, we instead use the Jacboian matrices of the state equations.
The LQR should return a controller

$$u_t = -Ks_t, \tag{A.33}$$

that minimizes the quadratic cost

$$J = \sum_t \left(s_t^T W_1 s_t + u_t^T W_2 u_t\right), \tag{A.34}$$

where Q, R are the weight matrices of the cost functions. Here we assume that applying controls does not generate any cost and set $W_2 = 0$. The $W_1$ weight matrix is modeled so that $s_t^T W_1 s_t = H$ and therefore the quadratic cost represents the energy of the system.
Using this approach we get the following weight matrix $W_1$ for the inverted quadratic potential,

$$W_{\text{quadratic}} = \begin{bmatrix} \frac{k}{2} & 0 \\ 0 & \frac{1}{2m} \end{bmatrix}. \tag{A.35}$$

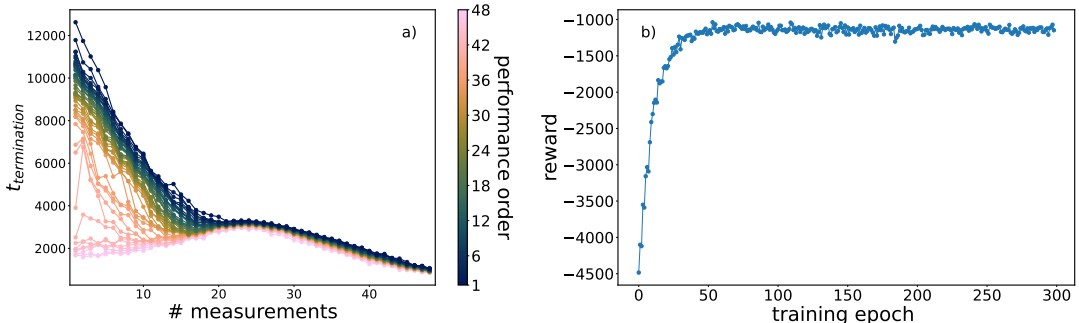

Figure 8: **Visualization of the training results** for training agents on the classical system with inverse quartic potential. In a) the average termination time of all agents after the training is shown against the number of measurements performed. For every measurement number 48 different agent are shown, and are color coded depending to rank their performance. In b) the process of training a RLE agent is showcased, where the reward are plotted against the first 300 training epochs.

In case that we have a nonlinear system, we approximate the dynamics according to Eq. A.29 and set the weights to be $s_t^T W_1 s_t = H|_{s_t}$:

$$W_{\text{cosine}} = \begin{bmatrix} \frac{k_1(\cos(\pi \hat{x}/k_2)-1)}{s^2} & 0 \\ 0 & \frac{1}{2m} \end{bmatrix}, \qquad W_{\text{quartic}} = \begin{bmatrix} ks^2 & 0 \\ 0 & \frac{1}{2m} \end{bmatrix}, \tag{A.36}$$

where we evaluate the $W_{\text{cosine}}$ and $W_{\text{quart}}$ every timestep, according to the latest state estimation $\hat{s}_t$.

## A.5 Reinforcement learning training

The PPO algorithm trains a policy $\pi_\theta$ from which the actions $a$ are sampled allowing for exploration in the training. The policy is updated by maximizing the objective function $L$

$$\theta_{k+1} = \text{argmax}_\theta E_{s,a \approx \pi_{\theta_k}} \left[ L(s, a, \theta, \theta_k) \right], \tag{A.37}$$

with $L$:

$$L(s, a, \theta, \theta_k) = \min \left( \frac{\pi_\theta(a,s)}{\pi_{\theta_k}(a,s)} A^{\pi_{\theta_k}}(s,a), g(\epsilon, A^{\pi_{\theta_k}}(s,a)) \right). \tag{A.38}$$

Since reinforcement learning is known to be vulnerable to performance collapse [34, 35], caused by a few unfortunate episodes in the training, the PPO limits how far any new policy is allowed to differ from the previous one, by clipping the probability ratio $\frac{\pi_\theta(a,s)}{\pi_{\theta_k}(a,s)}$ in the objective function

$$g(\epsilon, A) = \begin{cases} (1+\epsilon)A, & A \geq 0, \\ (1-\epsilon)A, & A < 0, \end{cases} \tag{A.39}$$

with $\epsilon$ controlling the clipping range. The new policy does not benefit by going far away from the old policy.

In the training of the reinforcement learning models, we used 2 different sets of hyperparameters. One for the training of the RLC models and one for the RLE.

Table 2: **Hyperparameters of the reinforcement learning approach**, specifying the initialization of the RLC and RLE agents and their respective training process.

| Reinforcement Learning Parameters | | |
|---|---|---|
| Parameter | RLE | RLC |
| learning rate | 2e-05 $\cdot N_{\text{meas}}$ | 3e-05 |
| clipping rate | 0.7 | 0.5 |
| epochs | 1000 | 10000 |
| steps per epoch | 100000 | 200000 |
| batchsize | 2048 | 1024 |
| nepochs | 20 | 10 |
| action network | [32, 32] | [32, 32] |
| value network | [32, 32] | [32, 32] |
| activation function | tanh | tanh |

The training of the RLE and RLC agents is done by utilizing the same underlying methods. There is a variance in the used hyperparameters, which were determined using grid search and are listed in the Tab. 2. During the training we tracked the returned reward after each epoch and saved the models, which returned the highest reward.

The training of the RLC agent turned out be easily affected by getting stuck in local minima, completely halting the learning process. Since this more often happened at small number of weak measurements, we have utilized transfer learning [36] to circumvent this problem. In transfer learning we trained 48 agents on $N_{\text{meas}} = 48$ weak measurements. These agents were then used as the starting point for training with $N_{\text{meas}} = 47$ weak measurements. This was repeated for each number of weak measurements until $N_{\text{meas}} = 1$ is reached. The trained agents have then been evaluated on the potential as shown in 8 a) for the Kalman FIlter + RLC controller on a classical system with an inverted quartic potential. At the starting point of the training at $N_{\text{meas}} = 48$, the agents all show a performance close to on another, but as the transfer learning continues, at around $N_{\text{meas}} = 20$, the performance of the agents starts to diverge. The majority of the agents exhibits increasing performance with the number of measurements, while a small number of agents get stuck during the training and only show a

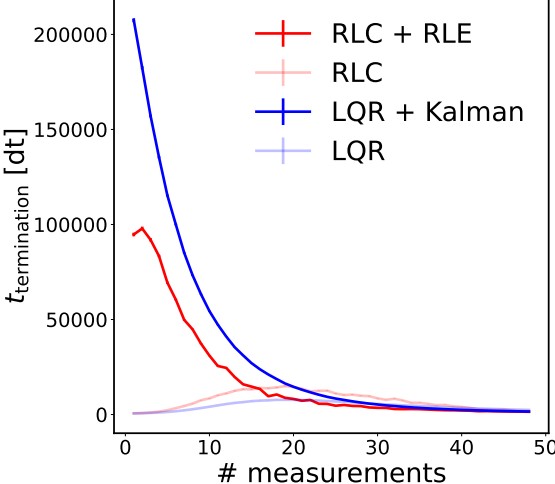

Figure 9: **Comparison of RL vs LQR & Kalman Filter** on the classical cartpole (with inverse quadratic system). The noise level is fixed at $\sigma_{\text{meas}} = 0.5$ and state estimators in form of the Kalman Filter and RLE are added towards the comparison.

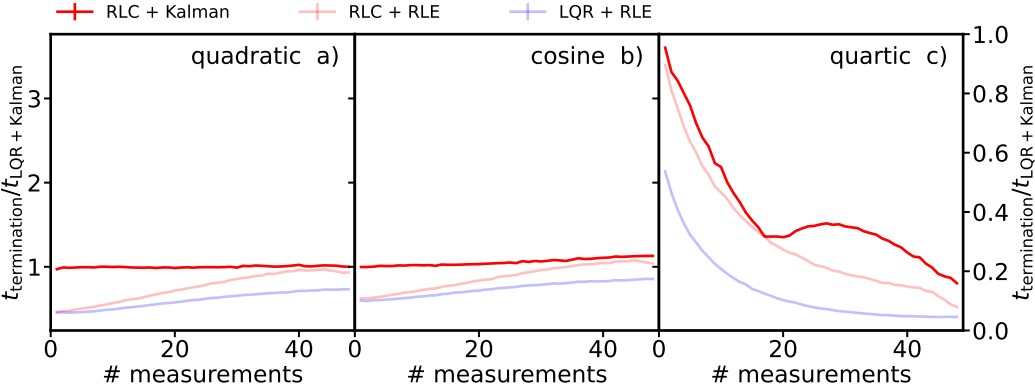

Figure 10: **Benchmarks of the various controllers**, on the **classical** system depending on the number of measurements for the input. We showcase the performance of classical controller using Kalman Filter + LQR (blue) and the pure reinforcement learning controller (light red), as well as the mix of classical and reinforcement learning controllers with Kalman Filter + RLC (red) and RLE + LQR (light blue). The performance is showcased as ratio of the average termination time between a selected controller and the Kalman + LQR controller. Each plot represents a different potential, the first being the **quadratic** potential, followed by the **cosine** potential and **quartic** potential.

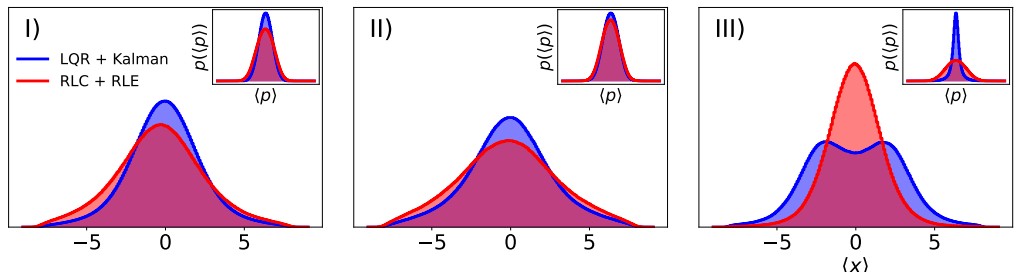

Figure 11: **Distributions of the position and momentum of the stabilized wavepacket** on the **classical** system. The position $\langle x \rangle$ of the wavefunction was tracked over $10^6$ timesteps for the **LQR + Kalman** controller (blue) as well as the **RLC + RLE** (red) controller and converted to histogram of the position probability $p(\langle x \rangle)$. This is also showcased for the momentum $\langle p \rangle$ in the top right inserts. The three plots represent measurement from different potentials, those being the **quadratic** (I), **cosine** (II) and **quartic** (III) potential.

fraction of the performance of other agents. THis shows the difficulty of the training process. Because the training of the RLE isn't directly depending on the average termination time of the wavefunction, we don't need to make use of transfer learning and can train the 48 agents directly, independent from each other, one agent for every number of measurements we perform. In Fig. 8 b) the training process of the RLE agent for $N_{meas} = 1$ in the classical system with inverse quartic potential is shown, demonstrating a fast converging in the training.

## A.6 Extended classical benchmarks

In Fig. 9 the performance of adding an estimator to the RLC and LQR is showcased. Compared to Fig. 3 the noise level is fixed at $\sigma_{meas} = 0.7$, resulting in higher overall performance for all controller. Furthermore the performance gap has widened while using an estimator, with RLC

+ RLE achieving a higher peak performance by a factor of $\approx 5$ compared to the RLC controller.

Performing the extended benchmarks on the classical surrogate model yields results similar to the corresponding quantum test. Fig. 10 illustrates that, once again, the Kalman + RLC model matches the performance of the LQGC for the quadratic and cosine potential. Only in the case of the quartic potential is a performance advantage observed, with the performance increasing by a factor $\geq 3$. Those results showcase a greater performance increase than that on the quantum environment.

The controller combinations involving the RLE also only show a performance advantage in the quartic potential. Compared to the quantum version, here the pure RL controller (RLC + RLE) is able the showcase almost the same performance as RLC + Kalman.

Looking closer at the controller behaviour in the classical case, one can see in Fig. 11, that the general observations from Fig. 4 remain valid. The differences are that the position distribution of the RL controller in the cosine potential now displays a distinct peak around the center of the potential, instead of having a plateau. Moreover, the position distribution of the LQGC controller in the quartic case actually loses its single peak and is replaced by a symmetrical double peak located around the center of the potential.

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
