# Peer review of "The Quantum Cartpole: A benchmark environment for non-linear reinforcement learning"

_SciPost Physics Core, doi:SciPost Phys. Core 7, 026 (2024)_

## Round 1 · Referee Report · Anonymous · 2024-2-7

Strengths
Extensive analysis of the quantum cartpole stabilization using reinforcement learning
Weaknesses
Technical language
Missing information in main text on assessment criteria
Report
The paper is an interesting and detailed numerical study of the efficiency of a feedback-based control algorithm. The study is performed for the dynamics of the classical and quantum cartpole, with and without state estimator and for different potentials.
The language is quite technical, the authors use several acronyms and remaind the reader to the appendix - also for important definitions without which the text results hard to understand.
The authors do not explain the criteria they employ for assessing the performance. This concerns Fig. 3, 4, and 6 (Here t_termination, not defined, appears on another unspecified scale, different from the one of Fig. 3 and 4). I could not find any explanation of the white line in Fig. 4.
This paper could be published in some form after the authors have revised the text. In view of the acceptance criteria of SciPost Physics (https://scipost.org/SciPostPhys/about#criteria) I cannot recommend this work: the advance with respect to the existing paper of Wang et al (Ref. [7] of this paper) is majorly technical. On the other hand, the work here presented is useful and contains interesting ideas for future works. I therefore recommend to consider the paper for publication in SciPost Physics Core.
Requested changes
See report

---

## Round 1 · Referee Report · Anonymous · 2024-2-16

Strengths
1- The results seams to be correct and the code is available
2- Comprehensive comparison between LQGC and PPO on the quantum cartpole and it's quantum version.
Weaknesses
1- Results not too impactful, given already Ref.[7].
2- Some sections are not clearly written and some important details are missing.
Report
The authors consider the quantum cartpole problem and it's classical analog (which is not the classical cartpole problem). They apply the LQGC algorithm and the PPO (a Reinforcement Learning algorithm) with and without an estimator (the Kalman filter and an other PPO agent) in order to solve these problems (don't let the quantum or classical particle fall down the hill).
I do not recommend publication of this manuscript in SciPost Physics because it does not fulfill the acceptance criteria, in particular the expectations criteria: I think the system is quite simple and I do not think the methods and the results are too novel, given also ref.[7]. However the results are correct and interesting and I suggest publication in SciPost Physics Core.
Requested changes
1) The explanation of the quantum cartpole in not clear:
1a) at the beginning of page 4 they use $dt$ and $\Delta t$ and it is not clear if it is a typo or if they are different. Related to this, it is not clear if, when changing $N_{meas}$, the time between each measurement is proportionally varied (and thus the time between each control is constant) or if it is kept constant (and thus the time between each control changes).
2) It is not clear how the estimator agent is then used in the control agent. This detail is very important in order to have a fair comparison between the performance of the control agents with and without state estimators. In fact, it is important to clarify that the "control agent + estimator agent" do not have access to more information about the system than the "control agent" alone. Otherwise the improvement in performance is obvious.

---

## Round 2 · Author Response

Referee 1:
Comment: The authors do not explain the criteria they employ for assessing the performance. This concerns Fig. 3, 4, and 6 (Here t_termination, not defined, appears on another unspecified scale, different from the one of Fig. 3 and 4). I could not find any explanation of the white line in Fig. 4.
Reply: We have added a detailed description of t_termination on page 5 and added an explanation of the white line in the caption of Fig. 4. For Fig. 6 we have chosen a relative scale weighing the different controllers against the LQGC to highlight the comparative performance, independent from the maximum performance depending on the underlying potential. We have added this explanation for the chosen scale on p.8.
Referee 2:
Comment: At the beginning of page 4 they use dt and Δt and it is not clear if it is a typo or if they are different. Related to this, it is not clear if, when changing Nmeas, the time between each measurement is proportionally varied (and thus the time between each control is constant) or if it is kept constant (and thus the time between each control changes)
Reply: We have fixed the typo on page 4 and changed dt to Δt. Also we have extended the explanation of N_meas, clarifying that Δt remains constant, even when increasing N_meas.
Comment: It is not clear how the estimator agent is then used in the control agent. This detail is very important in order to have a fair comparison between the performance of the control agents with and without state estimators. In fact, it is important to clarify that the "control agent + estimator agent" do not have access to more information about the system than the "control agent" alone. Otherwise the improvement in performance is obvious.
Reply: The difference in information between the RLC and the RLE is that the RLE additionally uses the previously estimated state and the control force used. The difference in the information provided is the same as for the LQR compared to the Kalman filter. The subsequent increase in performance shown in Fig. 3b) can therefore be expected for N_meas = 1. For N_meas > 1 it should also be noted the framestacking of the measurement has to be taken into consideration. Therefore, the figure also highlights the interaction between the simultaneous use of estimators and framestacking, and shows that framestacking plus a simple controller could already be used as a simple quantum control technique. Additionally, Fig. 3 shows that learning the estimator model is more likely to be the bottleneck, as the RLC can exhibit the same or better performance than the LQR. We have expanded the interpretation of Fig. 3 on p.7.

---

## Round 2 · List of Changes

1. We have added the meaning of the white line in Fig. 4 to the caption of the figure.
2. In the middle of page 8, we have added an explanation of the scale chosen for Fig. 6.
3. Changed a typo from dt to Δt at the beginning of page 4.
4.In the middle of page 4, we have expanded the description of the quantum cartpole setup to clarify that Δt remains constant.
5. We have extended the interpretation of the results of Fig. 3 on page 7.

---

## Editorial Decision

published